Multimodal Imaging Brain Connectivity Analysis (MIBCA) toolbox

Ribeiro Andre Santos 1 3 afs13@imperial.ac.uk
Lacerda Luis Miguel 2 3
Ferreira Hugo Alexandre 3
1 Division of Brain Sciences, Department of Medicine, Centre for Neuropsychopharmacology, Imperial College London , UK
2 Department of Neuroimaging, Institute of Psychiatry, Psychology and Neuroscience, King’s College London , UK
3 Faculty of Sciences, Institute of Biophysics and Biomedical Engineering, University of Lisbon , Portugal
Tavano Alessandro
Electronic publication date: 2015 Jul 14
Publication date: 2015
Volume: 3
Electronic Location ID: e1078
Received 2014 Dec 16; Accepted 2015 Jun 14
Copyright: © 2015 Ribeiro et al.
Copyright year: 2015
Copyright holder: Ribeiro et al.
License: This is an open access article distributed under the terms of the Creative Commons Attribution License, which permits unrestricted use, distribution, reproduction and adaptation in any medium and for any purpose provided that it is properly attributed. For attribution, the original author(s), title, publication source (PeerJ) and either DOI or URL of the article must be cited.
License URL: https://creativecommons.org/licenses/by/4.0/

Keywords: DTI, MRI, Toolbox, Connectivity, fMRI, PET, MIBCA, Multimodal

Funding: Fundação para a Ciência e Tecnologia (FCT) Ministério da Ciência e Educação (MCE) Portugal (PIDDAC) PTDC/SAU–ENB/120718/2010 This work was financed by Fundação para a Ciência e Tecnologia (FCT) and Ministério da Ciência e Educação (MCE) Portugal (PIDDAC) under grants PTDC/SAU–ENB/120718/2010 and PEst–OE/SAU/UI0645/2014. The funders had no role in study design, data collection and analysis, decision to publish, or preparation of the manuscript.

==============================
Aim. In recent years, connectivity studies using neuroimaging data have increased the understanding of the organization of large-scale structural and functional brain networks. However, data analysis is time consuming as rigorous procedures must be assured, from structuring data and pre-processing to modality specific data procedures. Until now, no single toolbox was able to perform such investigations on truly multimodal image data from beginning to end, including the combination of different connectivity analyses. Thus, we have developed the Multimodal Imaging Brain Connectivity Analysis (MIBCA) toolbox with the goal of diminishing time waste in data processing and to allow an innovative and comprehensive approach to brain connectivity.

Materials and Methods. The MIBCA toolbox is a fully automated all-in-one connectivity toolbox that offers pre-processing, connectivity and graph theoretical analyses of multimodal image data such as diffusion-weighted imaging, functional magnetic resonance imaging (fMRI) and positron emission tomography (PET). It was developed in MATLAB environment and pipelines well-known neuroimaging softwares such as Freesurfer, SPM, FSL, and Diffusion Toolkit. It further implements routines for the construction of structural, functional and effective or combined connectivity matrices, as well as, routines for the extraction and calculation of imaging and graph-theory metrics, the latter using also functions from the Brain Connectivity Toolbox. Finally, the toolbox performs group statistical analysis and enables data visualization in the form of matrices, 3D brain graphs and connectograms. In this paper the MIBCA toolbox is presented by illustrating its capabilities using multimodal image data from a group of 35 healthy subjects (19–73 years old) with volumetric T1-weighted, diffusion tensor imaging, and resting state fMRI data, and 10 subjets with 18F-Altanserin PET data also.

Results. It was observed both a high inter-hemispheric symmetry and an intra-hemispheric modularity associated with structural data, whilst functional data presented lower inter-hemispheric symmetry and a high inter-hemispheric modularity. Furthermore, when testing for differences between two subgroups (<40 and >40 years old adults) we observed a significant reduction in the volume and thickness, and an increase in the mean diffusivity of most of the subcortical/cortical regions.

Conclusion. While bridging the gap between the high numbers of packages and tools widely available for the neuroimaging community in one toolbox, MIBCA also offers different possibilities for combining, analysing and visualising data in novel ways, enabling a better understanding of the human brain.

Introduction

For a long time there has been an interest in unravelling the mechanisms and circuitry that allow human beings to perform very complex tasks. In particular, the study of how the brain is organized and how different brain regions communicate (brain connectivity) has been thoroughly studied at several scales. At the microscale, the first techniques were developed for the measurement of electrical activity in animals (Hodgkin & Huxley, 1952), which allowed for individual neuronal communication and synaptic activity to be detected. Another approach was to perform post-mortem dissections of neural tissue and try to infer the architecture of different neuro-anatomical systems (Buren & Baldwin, 1958). Histological staining was used to distinguish between different types of neurons (the functional and structural unit of the central nervous system) (Eickhoff et al., 2006). The first results were obtained by the use of techniques such as animal axonal tracing, which allows to undercover the neural connections from its origin to where they project (Schmahmann et al., 2007). However, these studies are unable to show how the structure is linked to individual functions and are generally of invasive nature or inapplicable to humans. Therefore, the motivation of the study of macroscale brain connectivity, which is much more accessible for human studies, is to map different patterns of activation and different routes of information that link highly specialized centres of information and explain their integration in the major network (Rubinov & Sporns, 2010).

Macroscale brain connectivity

Macroscale brain connectivity is generally assessed at various different modes or types, but it can be mainly subdivided into structural, functional, and effective connectivity (Fig. 1). Structural connectivity is linked to the routes of information in the brain and how they allow information to be transmitted. It can be measured using diffusion-weighted magnetic resonance imaging (dMRI) where the displacement of water molecules is used to trace a three dimensional reconstruction of their path in the brain via tractography (Catani et al., 2013). Tractography is not only extremely useful to localize tracts on a subject, but also to register tracts into an atlas, and to understand or predict dysfunction caused by (structural) disconnections in specific locations (Catani & Mesulam, 2008), remaining the only tool to do so in vivo and non-invasively. Additionally, it is also very important in the study of brain connectivity, even with limited spatial resolution (at the scale of millimetres) (Jbabdi & Johansen-Berg, 2011). On the other hand, functional connectivity, on the other hand, demonstrates how different areas of the brain with similar patterns of activation enable brain functions at rest and in response to external stimuli (Van den Heuvel et al., 2009). It has helped undercover concepts about the basal level of activations in the brain as reflected in the more commonly described resting state networks, of which the default mode network has been one of the more exploited (Behrens & Sporns, 2012). Functional magnetic resonance (fMRI) is one of the tools that has provided such information by inferring changes in the local magnetic properties of blood in response to variable brain activity (Ogawa et al., 1990). Functional metabolic changes have also been explored with Positron emission tomography (PET), where radioactive tracers are injected in the body and bind to target molecules of interests to measure their activity over time (Friston et al., 1993). In addition to these techniques, there are a family of methods that allow the exploration of the electric properties of neuronal conduction, whether by measuring them directly using electrodes such as Electroencephalography (EEG) (Berger, 1933) or the magnetic fields generated by them, using Magnetoencephalography (MEG) (Cohen, 1968). Finally, effective connectivity may be seen as a way of combining the two types of connectivity described above, where the intention is to infer a causal relation between functionally linked activated areas and how they can be related through structural connections depicted independently (Frye, 2011).

Different types of analysis have been reported for brain connectivity studies, and a large interest has arisen in the field of network theory and connectomics (Sporns, Tononi & Edelman, 2000), in which connectivity metrics are extracted from functional and structural neuroimaging techniques. These techniques assume that the information collected from neuroimaging data, representing different aspects of brain anatomy and function, can be encoded as a graph (Ginestet et al., 2011). A graph is a collection of nodes linked with each other via edges and can, therefore, represent different regions of the brain and the interplay of information between them. Depending on the type of information, one can have undirected or directed graphs (if the information holds no directionality or entails some sort of casual response (Bassett & Bullmore, 2006), respectively) for structural/functional and effective connectivity (Fig. 1).

Figure 1 Macroscopic brain connectivity display through 3D Graphs.

(A) Undirect and weighted Structural Connectivity (different number of connections). (B) Undirect and weighted Functional Connectivity (different node strength connexion). (C) Direct and weighted Effective Connectivity (different directionality connections).

Brain connectivity analysis toolboxes

There are several available toolboxes that use a single and (to some extent) more than one neuroimaging technique to perform individual connectivity analysis but not in a truly multimodal fashion, where information is combined from the very beginning to the desired goal by the combination of different connectivity analyses (Cui et al., 2013; Zhou, Thompson & Siegle, 2009; Song et al., 2011; Chao-Gan & Yu-Feng, 2010; Zang et al., 2012; Lei et al., 2011a; Lei et al., 2011b; He et al., 2011; Whitfield-Gabrieli & Nieto-Castanon, 2012; Seth, 2010; Hadi Hosseini, Hoeft & Kesler, 2012; Marques et al., 2013). In this work, we summarize the different toolboxes divided into Structural, Functional, Effective and Multimodal.

Structural connectivity toolboxes

Pipeline for Analysing braiN Diffusion imAges (PANDA) (Cui et al., 2013): This toolbox allows fully automated processing of brain diffusion images. The tool uses processing modules of established packages, including FMRIB Software Library (FSL Smith et al., 2004), Pipeline System for Octave and Matlab (The MathWorks Inc., Natick, Massachusetts, USA, 2000), PSOM (https://code.google.com/p/psom/), Diffusion Toolkit (Wang et al., 2007) and MRIcron (Rorden, Karnath & Bonilha, 2007). Using any number of raw dMRI datasets from different subjects, in either DICOM or NIfTI format, PANDA can automatically perform a series of steps to process DICOM/NIfTI and extract metrics with the diffusion tensor imaging (DTI) formalism that are ready for statistical analysis at the voxel-level, the atlas-level and the Tract-Based Spatial Statistics (TBSS)-level.

Functional connectivity toolboxes

MATLAB toolbox for functional connectivity (Zhou, Thompson & Siegle, 2009): This toolbox calculates functional connectivity measures extracted from both resting state functional magnetic resonance imaging (rs-fMRI) and task based blood oxygen level dependent (BOLD) data, using a free and user-friendly interface available through MATLAB. These measures are categorized into two groups: whole time-series and trial-based approaches, including zero-order and cross-correlation, cross-coherence, mutual information, peak correlation, and functional canonical correlation.

RESting-state fMRI data analysis Toolkit (REST) (Song et al., 2011): Based on MATLAB, REST can exchange files/data with SPM, AFNI, and FSL under the NIfTI or ANALYZE formats. After data preprocessing with SPM or AFNI, a few analytic methods can be performed in REST, including functional connectivity analysis based on linear correlation, regional homogeneity, amplitude of low frequency fluctuation (ALFF), and fractional ALFF. To increase the processing capability and usability of REST, an extension was implemented: Data Processing Assistant for Resting-State fMRI (DPARSF) (Chao-Gan & Yu-Feng, 2010).

Conn (Whitfield-Gabrieli & Nieto-Castanon, 2012): Conn is able to spatially and temporally pre-process fMRI data as well as perform first- and second- level analysis. The toolbox also offers a batch processing environment facilitating the implementation of functional connectivity analysis. Conn is also able to derive Graph theory measures from fMRI measures.

Effective connectivity toolboxes

Multimodal Functional Network Connectivity (mFNC) (Lei et al., 2011a): The mFNC toolbox was proposed for the fusion of Electroencephalography (EEG) and fMRI in network space. First, functional networks (FNs) are extracted using spatial independent component analysis (ICA) separately in each modality. Then, the interactions among FNs in each modality are explored by Granger causality (GC) analysis. The fMRI FNs are then matched to EEG FNs in the spatial domain using network based source imaging (NESOI Lei et al., 2011b).

Electrophysiological Connectome (eConnectome) (He et al., 2011): Major functions of eConnectome include EEG and Electrocorticography (ECoG) preprocessing, scalp spatial mapping, cortical source estimation, connectivity analysis, and visualization.

Granger causal connectivity analysis (GCCA) (Seth, 2010): The GCCA toolbox provides a range of MATLAB functions (but without a Graphical User Interface (GUI)) enabling the application of Granger-causality analysis to a broad range of neuroscience data.

REST-Granger causality analysis (REST-GCA) (Zang et al., 2012): REST-GCA is the second extension of the REST toolbox. It integrates two algorithms of GCA and provides a transformation programme of residual-based F to normal-distributed Z scores.

Structural-functional connectivity toolboxes

Connectome Mapping Toolkit (CMTK) (Gerhard et al., 2011): The CMTK comprises the Connectome Mapper and the Connectome Viewer Toolkit software. Although the Connectome Mapper computes only structural connectivity matrices from dMRI registered to a T1-weighted atlas, the Connectome Viewer Toolkit (CVT) is able to visualize both dMRI connectivity and fMRI connectivity processed elsewhere. CVT has the particularity of having been developed in Python, and uses a number of available libraries for visualisation, graph metrics calculation and network analysis.

Graph-Theoretical Analysis Toolbox (GAT) (Hadi Hosseini, Hoeft & Kesler, 2012): GAT provides a GUI that facilitates construction and analysis of brain networks, comparison of regional and global topological properties between networks, analysis of network hub and modules, and analysis of resilience of the networks to random failure and targeted attack. GAT does not, however, provide a preprocessing pipeline for the different modalities and is not yet able to analyse weighted and directed networks.

The UCLA Multimodal Connectivity Database (UMCD) (Brown et al., 2012): The UMCD is a web platform for connectivity matrix data repository, sharing and analysis. The platform is able to analyse connectivity matrices derived from imaging techniques such as DTI or rs-fMRI using graph theory, and builds a report of the data. Like the previous toolbox, the platform does not provide a preprocessing pipeline.

BrainNet Viewer (BNV) (Xia, Wang & He, 2013): The BNV is another toolbox for visualization only. It uses connectivity matrices computed elsewhere and has a number of options for displaying connectivity data and graphs.

BrainCAT (Marques et al., 2013): BrainCAT implements a predefined pipeline for fMRI and DTI data preprocessing, ICA of the fMRI data and combination with DTI tractography analysis. BrainCAT does not yet apply connectivity analysis such as graph theory, nor does it easily allow the combination of different modalities in more complex ways.

Connectome Visualization Utility (CVU) (LaPlante et al., 2014): CVU is a visualization software for connectivity analysis that includes matrix visualization, Connectogram view, and 3D Graph view for different modalities. However, the visualization features of CVU are limited to single-edged, undirected networks, i.e., effective connectivity cannot be visualized through this software. Further, CVU is unable to represent networks from multiple modalities in the same file.

The Virtual Brain (TVB) empirical data pipeline (Schirner et al., 2015): This pipeline derives structural and functional connectivity data from dMRI and fMRI, and optionally from EEG. Also, this pipeline has the particularity of being able to compute a novel structural connectivity metric to quantify the strength of signal transmission between regions. The computed connectivity data can then be imported to the TVB, a brain simulation platform that also displays connectivity relations in the form of matrices, graphs and topography maps (Sanz Leon et al., 2013).

Anatomical-structural-functional-effective connectivity toolbox

In this paper, we propose a fully automated all-in-one connectivity analysis toolbox—Multimodal Imaging Brain Connectivity Analysis toolbox (MIBCA)—that offers pre-processing, connectivity and graph theoretical analyses of multimodal image data such as anatomical MRI (aMRI), dMRI, fMRI and Positron Emission Tomography (PET). MIBCA was developed as an effort to diminish research time waste by pipelining state-of-the-art methods and to allow an innovative approach to brain connectivity research via multimodal matrix analysis and graph theory metrics. It has been shown before that knowledge derived from structural connectivity is important to recover functional aspects of the brain, such as predicting the degree of neural dynamics based on anatomical patterns variability (Honey, Thivierge & Sporns, 2010). Furthermore, the combination of both types of information may lead to a more accurate depiction of how the brain works, providing care is taken into account for the specifications of each technique. Specifically, tractography can be used to target the placement of electrodes for MEG/EEG for improved spatial localization of electric activity and also to drive better functional connectivity analyses (Phillips et al., 2012). Additionally, the recent development of integrated systems such as PET-MR scanners is expected to bring novel highlights into multimodal connectivity analysis (Shah et al., 2013), systems for which the MIBCA toolbox is particularly suited.

In particular, the MIBCA toolbox is able to combine information from the different connectivity matrices in order to enable novel insights into the data, such as the depiction of directed and mediated connections in the brain, as shown later in the article. A comparison between the different toolboxes is presented in Table 1.

In this paper we present the MIBCA toolbox and several of its applications. The software, tutorial and test data is made available upon registration in http://www.mibca.com/.1 The software is free to download and follows the Creative Commons Copyright License (CC-BY) 4.0.

Table 1 Comparison between different connectivity toolboxes.

Toolbox	Pre- Process.	Anat. Conn. MRI	Struct. Conn. MRI	Funct. Conn. MRI	Effect. Conn. MRI	Funct. Conn. PET	Funct. Conn. EEG	Imaging/ Physio.	Graph Analy.	Conn. Visual	Group Stats	References	
PANDA	X		X					X	X			Cui et al., 2013	
FCT				X				X			X	Zhou, Thompson & Siegle, 2009	
DPARSF/ REST/ REST-GCA	X			X				X		X	X	Chao-Gan & Yu-Feng, 2010; Song et al., 2011; Zang et al., 2012	
Conn	X			X					X		X	Whitfield-Gabrieli & Nieto-Castanon, 2012	
mFNC	X			X	X		X					Lei et al., 2011a; Lei et al., 2011b	
eConnectome	X				X		X	X		X		He et al., 2011	
GCCA	X				X		X		X	X		Seth, 2010	
CMTK	X		X	X			X	X	X	X	X	Gerhard et al., 2011	
GAT		X	X	X					X	X	X	Hadi Hosseini, Hoeft & Kesler, 2012	
UMCD		X	X	X					X	X	X	Brown et al., 2012	
BNV		X	X	X	X					X		Xia, Wang & He, 2013	
BrainCat	X		X	X						X		Marques et al., 2013	
CVU			X	X					X	X		LaPlante et al., 2014	
TVB data pipeline	X		X	X			X		X	X		Schirner et al., 2015	
MIBCA	X	X	X	X	X	X		X	X	X	X	Ribeiro et al., 2015 (This article)	
Notes.

Pre-Process. Pre-processing of raw data

Anat. Conn. MRI Anatomical connectivity derived from T1-weighted MRI data

Struct. Conn. MRI Structural connectivity derived from diffusion MRI data

Funct. Conn. MRI Functional connectivity derived from blood oxygen level dependent (BOLD) (functional MRI) data

Effect. Conn. MRI Effective connectivity derived from BOLD (functional MRI) data

Funct. Conn. PET Functional/Metabolic connectivity derived from PET data

Funct. Conn. EEG Functional/Effective connectivity derived from EEG, ECoG and/or MEG data

Physio. Physiological (EEG)

Graph Analy. Graph analysis

Conn. Visual. Connectivity visualization tools

Group Stats Statistical analysis/Classification of groups of subjects

See text for details.

Material and Methods

MIBCA

MIBCA is an application developed in MATLAB (Fig. 2) and combines multiple freely-available tools in order to optimize and automate data processing pipeline, and to combine different imaging modalities into the same framework.

Figure 2 Main interface of the MIBCA toolbox.

Currently, MIBCA’s framework is able to process aMRI from volumetric T1-weighted data, dMRI from DTI data, resting state or task-based fMRI, and PET. Succinctly, MIBCA is organized as follows: pre-processing of the data from the different modalities, connectivity matrix estimation and graph theory analysis, and visualization Fig. 3.

Figure 3 MIBCA processing pipeline diagram.

Filled circles, Data generated and used on this study; Non-filled circles, Data generated but not used on this study. Gray circle, Database; Green circles, pre-processing; Blue circles, Connectivity Matrix estimation; Purple circles, Graph Theory analysis; Red circles, Group analysis.

A default pipeline is recommended to perform a typical analysis, even though the user has the option to add or remove certain steps, as well as modifying the processing parameters for each step.

After technique/modality specific connectivity matrix computation, MIBCA enables matrix operations to generate new connectivity data (e.g., Structural + Functional) and also intra-modality and inter-modality group analysis. Finally, MIBCA allows the user to visualize the computed connectivity data in a matrix form (Fig. 4), 3D-graph (Fig. 5) and/or a connectogram (Fig. 9).

Figure 4 Matrix visualization of structural (A), (B) and (C), and functional (D), (E) and (F) brain connectivities.

(A) and (D) presented the mean connectivity matrices; (B) and (E) robustness connectivity matrices; (C) and (F) combined connectivity matrices. A jet color code was used for all matrices, leading to the representation of lower values with colder colors and higher values with warmer colors. Matrix A shows the matrix organization with rows/columns (I, left/right subcortical regions; II, left cortical regions; III, right cortical regions) and areas (I/I, intra-subcortical connections; II/II, intra-hemispheircal left connections; III/III, intra-hemispherical right connections; I/II and I/III, subcortical-cortical connections; II/III, inter-hemispherical connections). Matrix organization is symmetrical with respect to the principal diagonal.

Figure 5 3D Graph visualization of structural (A) and (C) and functional (B) and (D) brain connectivities.

Here we considered the respective robustness matrices data (A) and (B) and highlighted the highest degree brain regions, (C) and (D).

Preprocessing

For each subject, raw data (aMRI, dMRI, fMRI and PET) are automatically preprocessed using state-of-the-art processing toolboxes, namely Freesurfer (aMRI) (Fischl, 2012), Diffusion Toolkit (dMRI) (Wang et al., 2007), FSL (fMRI) (Smith et al., 2004) and PET (SPM) (Friston, 1995). Once processing has been completed, intra-modality and inter-modality group analysis is performed using processed data. A summary of the generated files is shown in Table 2.

Table 2 Generated files and folders description.

Due to the large amount of generated files during the pre-processing step, only the most relevant files are described here.

Filename	Contents	
..\[subjectName]\sMRI\aparc+aseg.nii	Cortical and sub-cortical atlas image registered to aMRI	
..\[subjectName]\sMRI\[subjectName]_smri_data.mat	Matlab file (.mat) with extracted aMRI connectivity metrics	
..\[subjectName]\sMRI\[run#]\[run#].nii.gz	NIFTI 3d raw T1 image	
..\[subjectName]\DTI\aparc+aseg2DTI.nii	Cortical and sub-cortical atlas image registered to dMRI	
..\[subjectName]\DTI\[subjectName]_dti_data.mat	Matlab file (.mat) with extracted dMRI connectivity metrics	
..\[subjectName]\DTI\[run#]\[run#].nii.gz	NIFTI 4d raw diffusion image	
..\[subjectName]\DTI\[run#]\[run#].bvec	gradient vector file	
..\[subjectName]\DTI\[run#]\[run#].bval	b-value file	
..\[subjectName]\DTI\[run#]\[run#].dt\	DIFFUSION TOOLKIT folder with intermediate files	
..\[subjectName]\fMRI\aparc+aseg2fMRI.nii	Cortical and sub-cortical atlas image registered to fMRI	
..\[subjectName]\fMRI\[subjectName]_fmri_data.mat	Matlab file (.mat) with extracted fMRI connectivity metrics	
..\[subjectName]\fMRI\[run#]\[run#].nii.gz	NIFTI 4d raw functional MR image	
..\[subjectName]\fMRI\[run#]\[run#].feat\	FEAT folder with intermediate files	
..\[subjectName]\fMRI\[run#]\[run#].ica\	MELODIC folder with intermediate files	
..\[subjectName]\PET\aparc+aseg2PET.nii	Cortical and sub-cortical atlas image registered to PET	
..\[subjectName]\PET\[subjectName]_pet_data.mat	Matlab file (.mat) with extracted PET connectivity metrics	
..\[subjectName]\PET\[run#]\[run#].nii.gz	NIFTI 4d raw PET image	
..\[subjectName]\PET\[run#]\[run#].pet\	Generated PET folder with intermediate files	

Preparing the data for analysis

In neuroimaging studies, different imaging techniques and/or modalities are often used to acquire data from several healthy subjects or patients, thus resulting in large datasets.

To reduce the manual burden of researchers in sorting such amount of data, the developed toolbox is able to identify and process data following two simple rules. First, data must be organized in the following way: “Study-Subject-Acquisition-Images.” This allows the toolbox to differently pre-process images acquired with different imaging techniques or modalities (aMRI, dMRI, fMRI, PET), and to combine them in subsequent steps (Fig. 3—top row). Further, this organization allows the toolbox to identify different subjects and to use this information to automatically estimate group connectivity metrics (e.g., mean and standard deviation), as well as to perform group statistical tests (Fig. 3—bottom row). Second, although it is not required from the user to use fixed names for each acquisition image (e.g., aMRI, dMRI, fMRI, PET), it is required that each subject’s name be consistent for each technique/modality.

Furthermore, to maximize automation, different types of images can serve as input to the toolbox, such as DICOM, NIfTI, Analyze and ECAT; thus, the user is not required to convert between the different formats prior to the processing pipeline.

aMRI preprocessing

Each subject’s anatomical image is first corrected for intensity non-uniformity using the Non-parametric Non-uniform intensity Normalization (N3) (Sled & Zijdenbos, 1998). Next, the volume is registered in the MNI305 atlas through an affine registration. Intensity normalization and skull stripping are then performed to improve further processing. Data is non-linearly registered to an average brain, and the brain is parcellated into cortical and subcortical structures according to an atlas. The parcellated regions-of-interest (ROIs) are then mapped to the subject’s native space. Data finally follow a pipeline to derive the cortical thickness (CT), surface area (SA) and gray matter volume (GMV) measures for the subject cortical ROIs and also the volume of subcortical structures. All of the above processes were implemented using Freesurfer (http://surfer.nmr.mgh.harvard.edu/). The generated measures are then loaded and converted into a MATLAB (.mat) file. Finally, anatomical connectivity matrices (a-CM) are computed from ratios of these measures between each pair of ROIs (CT, SA and GMV connectivity matrices).

dMRI preprocessing

If the raw diffusion images are in DICOM format, then they are converted into a NIFTI 4D image, a gradient vector file (.bvec) and a b-value file (.bval) using the dcm2nii function available in the MRIcron package (Rorden, Karnath & Bonilha, 2007). Otherwise, the 4D image, gradient vector and b-value files are searched in the directory for further analysis. The raw images are first corrected for motion and eddy currents using eddy_correct (available in FSL). The diffusion tensor is reconstructed from the corrected images using dti_recon (available in Diffusion Toolkit). Processing results include the main eigenvector maps, b0 and diffusion weighted images, the apparent diffusion coefficient (ADC), Mean Diffusivity (MD) and Fractional anisotropy (FA) maps. For fiber tracking purposes, the dti_tracking (Diffusion Toolkit) function was used with an interpolated streamline method of fixed step-length and a deterministic tractography algorithm. The generated track file is first smoothed with the spline_filter (Diffusion Toolkit) and then loaded into MATLAB.

The b0 image generated from the dti_recon is non-linearly registered to the aMRI and the aligned parcellated ROIs mapped back to the diffusion space. The loaded track file is used to calculate the number of fibers, mean fiber length and mean fiber orientation between pairs of ROIs, thus providing 3 different matrices. The matrix of the number of fibers is defined as the structural connectivity matrix (s-CM). Additionally, the MD and FA mean values for each region are calculated.

fMRI preprocessing

FMRI data processing is carried out using FEAT (FMRI Expert Analysis Tool) Version 6.00, part of FSL. The following pre-statistics processing is applied: motion correction using MCFLIRT (Jenkinson et al., 2002); slice-timing correction using Fourier-space time-series phase-shifting; non-brain removal using BET (Smith, 2002); spatial smoothing using a Gaussian kernel of FWHM 8 mm; intensity normalization of the entire 4D dataset by a single multiplicative factor; high pass temporal filtering (Gaussian-weighted least-squares straight line fitting, with sigma = 50.0 s). The fMRI data is then non-linearly registered to the aMRI and the aligned parcellated ROIs mapped back to the fMRI space. For each ROI, the functional (BOLD) time series is extracted to MATLAB for further analysis.

Functional connectivity analysis involves calculating the Pearson correlation between each ROIs’ BOLD time series. A correlation matrix is then obtained for every ROI pair combination. The correlation matrix or functional connectivity matrix (f-CM) is finally thresholded for a significance level of 5% and Bonferroni correction.

Effective connectivity metrics are further evaluated for each pair of ROIs through the pairwise implementation of the time domain Granger Causality (Granger, 1969). From the estimated effective connectivity metrics for each pair of ROIs, an effective connectivity matrix (e-CM) is calculated.

PET preprocessing

The original PET data is first converted from the original format to NIFTI 4D. The converted PET data is further corrected for motion using SPM and smoothed with a 8 mm Gaussian filter. The dynamic PET data is then summed into a NIFTI 3D image and non-linearly registered to the aMRI. The aligned parcellated ROIs are then mapped back to the PET space through the inverse transformation. For each ROI the dynamic PET series are extracted and its mean value per ROI is calculated. The Pearson correlation is then used to generate a correlation matrix between each pair of ROIs (PET connectivity matrix). Further, for the summed image, the mean standard uptake values (SUV) are calculated for each ROI.

Group connectivity and graph theory analysis

For each subject, a hybrid structural + functional connectivity matrix (sf-CM) is calculated resulting from the multiplication of s-CM and f-CM matrices (if raw data of both modalities are provided). Further, binary s-CM (number of fibers > 0), f-CM (p-values < 0.05, Bonferroni corrected), and e-CM (p-values < 0.05, Bonferroni corrected) are generated.

After computation of matrices and metrics has been performed for all subjects, mean connectivity (mean-CM) and robustness connectivity (robustness-CM) matrices are computed. The mean-CM result from averaging technique/modality or hybrid connectivity weighted matrices translating information regarding the strength of connections (number of fibers in dMRI, or correlation coefficient in fMRI). The robustness-CM results from the mean of the binary matrices, providing a measure of the robustness of each connection (e.g., a value of 0.1 in the robustness-CM states that only 10% of the subjects show connections between a certain ROI pair, while a value of 0.9 states that those connections are present in 90% of the subjects).

Group s-CM, f-CM and sf-CM are further evaluated regarding general graph-theory metrics, namely mean network degree, mean clustering coefficient, characteristic path length and small-worldness calculated using the Brain Connectivity Toolbox (BCT: https://sites.google.com/site/bctnet/ (Rubinov & Sporns, 2010)). Additionally, normalized indexes based on these general metrics were calculated for an easier comparison between the different graphs. These normalized indexes were then calculated by the ratio of the metrics and their variant obtained from 10 random graphs, which were generated by shuffling the data-driven graph while maintaining symmetry and mean degree.

Individual ROI graph theory metrics, such as node degree and clustering coefficient, were also calculated using the BCT toolbox and saved for further analysis.

Visualization and statistical analysis

For an enhanced comprehension of the connectivity matrices and metrics three different types of visualisations were implemented, a matrix visualization (Fig. 4), a 3D Graph view (Fig. 5–8), and a Connectogram view (Figs. 9 and 10).

Figure 4A depicts matrix organization: left/right subcortical brain regions are represented in rows/columns. I; left cortical regions are represented in rows/columns II and right cortical regions are represented in rows/columns III. Thus, the intra-subcortical connections are represented in matrix area I/I, the intra-hemispherical left cortical connections are represented in matrix area II/II and the intra-hemispherical right cortical connections are represented in matrix area III/III. The remaining matrix areas I/II, I/III represent subcortical-cortical connections and matrix area II/III represent inter-hemispherical cortical connections. The matrix organization is symmetrical with respect to the principal diagonal. In the matrix visualization, any CM can be visualized using the jet color scheme (colder colours referring to lower values, and warmer colours to higher values).

The connectogram may contain, but is not limited to, the following variables as rings: ROIs labels, SA, GMV, CT, aMRI node degree, aMRI cluster coefficient, MD, FA, dMRI node degree, dMRI cluster coefficient, fMRI node degree, fMRI cluster coefficient, PET SUVs. The different ROIs are connected through lines based in one of the processed connectivity matrices, and can be, but not limited to: a-CM, s-CM, f-CM, sf-CM, e-CM. Line colour mapping can be based in any directed matrix produced, or a negative/positive encoded matrix (Figs. 9 and 10). Further, the connectogram can be used to observe connected regions to a certain ROI, by moving the cursor over the selected ROI, which turns blue while connected ROIs turn green.

The implemented 3D-Graph presents the same information as the connectogram but in a 3D view of the brain. Therefore, the different ROIs are connected through lines based in any of the processed connectivity matrices. Line colour mapping is again based in any directed matrix produced, or a negative/positive encoded matrix. The information from rings presented in the connectogram is shown in the 3D-Graph as a text box when the selected node is highlighted. As well as the connectogram, the 3D graph view is interactive and can be used to access to where a specific ROI is connected.

MIBCA also provides group comparison for all the above mentioned metrics and can display the significance of those tests both in a conectogram (Fig. 10) or 3D graph. The automated group analysis is based in the mean and standard deviation of the different connectivity matrices, as well as in the summation of the binary connectivity matrices, obtained for the total subject database. MIBCA enables the selection of different groups according to the available data and allows regression of specific variables of interest, displaying in red significant increases and in blue decreases for the performed tests.

A comprehensive manual of the toolbox with a more detailed description of data processing and capabilities is made available in the application website http://www.mibca.com.

Healthy subject data

A group of 35 healthy subjects (16 males and 19 females) with an average age of 43 ± 17 years (range: 19–73 years old) were selected from the ICBM database (http://www.loni.usc.edu/ICBM/) (Mazziotta et al., 2001) containing T1-weighted MR images (aMRI), 3 DTI series and 3 resting state fMRI series. PET data was only available for 10 subjects (5 males and 5 females) with an average age of 26.6 ± 5.7 years. In total, 255 images per subject were automatically processed by the toolbox.

T1-weighted, DTI and rs-fMRI data were acquired using a Siemens 1.5T scanner. T1-weighted images were acquired with a TR = 22 ms and a TE = 9.2 ms, yielding a matrix volume of 256 × 256 × 256 and a pixel spacing of 1 × 1 × 1 mm3. The DTI images were acquired with a TR = 8,000 ms, TE = 94 ms, b = 0, 1,000 s/mm2 and 35 non-collinear diffusion-sensitising gradients directions, yielding a matrix volume of 96 × 96 × 2,100 and a pixel spacing of 2.5 × 2.5 × 2.5 mm3. The fMRI images were acquired at rest with a TR = 2,000 ms and a TE = 50 ms, yielding a matrix volume of 320 × 320 × 138 and a pixel spacing of 4 × 4 × 5.5 mm3.

PET images were acquired for each subject using a Siemens HR+ scanner and 18F-Altanserin radiopharmaceutical for 30 min. The images were corrected for attenuation (measured), scatter (simulated 3D) and source decay, and reconstructed using the Fourier transform. The final image yielded a matrix volume of 128 × 128 × 63 with 8 frames and a pixel spacing of 2.06 × 2.06 × 2.43 mm3.

Results

Modality specific connectivity analysis

In order to test the MIBCA toolbox, we first investigated the connectivity similarities between s-CM (DTI based) and f-CM (rs-fMRI based) in different subjects using mean-CM (Figs 4A and 4D) and robustness-CM (Figs 4B and 4E). Further, we combined these matrices, through the following expression: ∗_c = ∗_mean × (∗_robustness > 0.8) (∗ representing the type of connectivity matrix), to obtain a matrix that preserved the information from the mean matrix, while assuring that the results were different from zero in at least 80% of the subjects.

In Fig. 4A it can be observed that, regarding structural connectivity, there is an average higher number of intra-hemispherical connections than inter-hemispherical connections. Conversely, regarding functional connectivity, both intra-hemispherical and inter-hemispherical connections are observed (Fig. 4D). It can also be seen a higher variability in functional connectivity than in structural connectivity (Figs. 4B and 4E). In particular, structural connectivity displays some clusters of intra-hemispherical regions with highly conserved connections between subjects (hot colours). The combined mean × robustness matrices (Fig. 4C and 4F) show more clearly the structural and functional brain connectivity organization: a predominance of intra-hemispherical structural connections and the more distributed intra and inter-hemispherical functional connectivity. Both structural and functional connectivity though, seem to roughly show a left–right symmetry.

This same information can be perceived from the 3D graph representations of the robustness s-CM and f-CM (Fig. 5A, 5B and 5D, respectively). These graph representations also illustrate the ROIs with higher degree (i.e., establish structural or functional connections with a higher number of regions), as they are represented as larger nodes. In particular, for the s-CM (Fig. 5C) the highest degree ROIs are the superiorfrontal and rostramiddlefrontal gyri, whilst for f-CM (Fig. 5D) these are the lingual gyri and insula.

Hybrid structural-functional connectivity analysis

The structural and functional connectivity matrices were further combined to generate a direct sf-CM matrix and an indirect/mediated sf-CM, Fig. 6. The direct sf-CM represents connections that are present in both s-CM and f-CM and therefore represent a direct structural (“axonal”) connection between two functionally related regions. The mediated sf-CM represents the connections of the f-CM that do not have a direct “axonal” connection between them, yet are functionally correlated. Here, in mediated sf-CM, connections between regions are presumably mediated by a third region or more regions. The sf-CM is therefore the sum of the direct and mediated sf-CMs.

Figure 6 3D Graph visualization of direct (A) and (C), and mediated (B) and (D) hybrid structural-functional brain connectivities.

(A) and (B) Full display of the hybrid connectivity matrices; (C) and (D) example of a mediated connection: functional connectivity between the rostral middle frontal gyri (D) mediated by the structural connections with the superior frontal gyri (C).

It can be seen in Fig. 6, that the number of direct connections (Fig. 6A) is smaller than the mediated connections (Fig. 6B). Furthermore, most of the direct connections are observed to be intra-hemispherical and short-range (small distance between nodes/ROIs), whilst the mediated connections tend to be longer range, and both intra- and inter-hemispherical.

To access which regions may mediate two functionally connected regions which are not connected directly, an algorithm that makes use of both source and target regions estimate the possible paths between them. As an example, the functional connectivity of the rostral middle frontal gyri was evaluated. As it can be seen, although there is no direct connection between them (easily seen in Fig. 6A), there is a possible path consisting of 4 regions2 (Fig. 6C): right rostral middle frontal gyrus → right superior frontal → left superior frontal → left rostral middle frontal gyrus.

To further understand if such path is viable the modularity of both structural and functional connectivity matrices were analysed, see Fig. 7. As shown, both the rostral middle frontal gyrus and the superior frontal gyrus belong to the same functional network (f-CM green module), yet also belong to two similar structural intra-hemispherical modules (s-CM’s dark blue and light blue modules).

Figure 7 3D Graph visualization of modularity of structural connectivity (A) and (C), and functional connectivity (B) and (D) matrices.

Figure 8 3D Graph visualization of effective connectivity restricted to direct connections.

Black lines, bidirectional connections; Gradient lines, directional connections from red to blue.

Finally, the dynamics of this connection was analysed through Granger causality using an analysis of order 1. As can be seen in Fig. 8 the connection between the rostral middle frontal gyri seems to be of an order >1 consistant with the mediation from the superior frontal gyri. The connection between the rostral middle frontal gyri and the superior frontal gyri seems to be bidirectional. Also, it is interesting that both rostral middle frontal regions presented an effective connection with the respective parstriangularis regions.

Multimodal connectivity analysis

The intra-inter hemispherical behaviour presented in the previous section can also be easily and comprehensively observed in Fig. 9. Direct connections (red lines) connect mostly regions of the same hemisphere (between same shade of gray of the outer ring), while the mediated connections (blue lines) tend to be both intra and inter-hemispherical (between different shades of gray). Additionally, we can observe that contra-lateral regions tend to present similar metric values (i.e., similar mean diffusivity, fractional anisotropy, cortical volume, SUVs, fmri and dti node degree), supporting the idea of a symmetric brain. This is easily seen for the subcortical regions for all metrics.

Figure 9 Multimodal connectogram.

From the outer to the inner ring: Regions-of-Interest (ROIs); Mean Diffusivity (MD); Fractional Anisotropy (FA); ROI volume; 18F-Altanserin rSUV (cerebellum as the reference region); functional connectivity degree (fMRI correlations); structural connectivity degree (diffusion tensor imaging-based tractography). Blue fibers, Direct structural/functional connections; Red fibers, Mediated functional connections.

Connectivity analysis using statistical tests

In this subsection is presented another potentiality of the MIBCA toolbox. In addition to the individual visualization of connectograms and 3D graphs for single and multimodal connectivity analysis, MIBCA also allows the user to perform statistical tests, while controlling for some variables of interest. In order to test this feature, subject data was divided into two age groups: 15 young adults (<40 years old) and 20 old adults (>40 years old), respectively with mean age of 26 ± 6 and 56 ± 8 years old and age range of 19–37 and 42–73 years old. A t-test was performed to evaluate significant differences between both groups. Figure 10 displays significant differences between both groups in several brain regions, both at the cortical and the subcortical level. The most notorious differences are regional decreased cortical thicknesses and grey matter volumes, as well as increases in mean diffusivity in the older subjects’ group in comparison with the younger subjects’ group.

Figure 10 Statistical test connectogram between younger and older subjects.

From the outer to the inner rings: Regions-of-Interest (ROIs), mean diffusivity, ROI volume, cortical thickness, fmri degree. The lines represent tract connections between brain regions. Blue and red colors, respectively, represent increased and decreased metrics’ mean values or number of tracts in older subjects group vs. youger subjects group.

Discussion

Here the MIBCA toolbox was presented. To the knowledge of the authors, this is the first toolbox that pre-processes both MRI and PET data and computes anatomical (T1-weighted based), structural (dWI), functional and effective connectivities. This toolbox also computes both imaging and graph-theory metrics, performs group statistics and displays data in matrices, brain graphs and connectograms, making it the first all-in-one connectivity toolbox (see Table 1). In particular, the MIBCA toolbox enables the researcher to combine multimodal information in a straightforward way using connectivity matrices operations, and therefore enabling innovative research questions. Finally, the toolbox does batch processing for multiple subjects, meaning that the researcher needs only to place raw data within a specific folder architecture and all the pre-processing and imaging and graph-theory connectivity computations are made autonomously. This saves the researcher from wasting time in repetitive operations and assures the correct pipeline in data processing, which is an advantage over other toolboxes.

In order to demonstrate the general capabilities of the toolbox, an example dataset was pre-processed and analysed. In Fig. 4, two different views of brain connectivity were proposed via the use of mean and robustness matrices, both of which are specific outputs of the developed toolbox resulting from matrices operations. Such an approach allows researchers to study subjects’ similarity and variability for different pairwise relations. Thus, it is possible to study the pairwise connectivity mean and variance,3 such as the distribution of the number of connecting fibers (obtained from s-CM). Additionally, both mean and robustness matrices can be combined to increase the confidence of the results.

Figures 4, 5 and 9 show that there is a predominance of intra-hemispherical connections in the s-CM, whilst this is not the case in f-CM. These results suggest that the brain is organized to have direct communication within the same hemisphere with fewer direct connections between hemispheres and, consequently, with a higher mediated inter-hemispherical communication (Park & Friston, 2013). Further analysis of the direct and mediated sf-CM (Fig. 6) show that direct connections tend to be short, while mediated connections appear to be long. These results also suggest that long-range connections can be based in more than one combination of direct short connections.

From these results, a mismatch of high degree nodes can be observed between the s-CM and f-CM (Fig. 5). For the s-CM, the higher degree nodes are related to regions where several physical connections to different cortical regions exist. For the f-CM, the high degree nodes are related to a high number of connections in resting-state. Therefore, although the first system is approximately static, the second is dynamic and can change when presented with a perturbation, probably leading to the observation of new higher degree nodes.

Moreover, in Figs. 4, 5, 7 and 8, a high regularity and hemispherical symmetry is shown for the s-CM, and in Fig. 9 a general symmetry for the different analysed metrics can be observed. For some systems in the human brain, this behaviour may be related to parallel computational processing in which two general processing units (the two hemispheres) process information simultaneously exchanging small packages of information between them, similarly to mechanisms present in visual processing (Baird et al., 2005; Doron & Gazzaniga, 2008). Additionally, the results suggest a certain degree of functional hemispherical asymmetry which may be related to the distributed behaviour of the brain, something that has been demonstrated in language pathways (Toga & Thompson, 2003; Catani et al., 2007).

Finally, an application of the MIBCA toolbox for the study of ageing in a multimodal approach was demonstrated (Fig. 10). The fairly generalized decrease in cortical thickness and regional gray matter volume, as well as the increased mean diffusivity observed in the group of older subjects, are consistent with neuronal loss commonly observed in healthy ageing (Salat et al., 2004; Minati, Grisoli & Bruzzone, 2007; Walhovd et al., 2005). Here, the statistical connectogram related in the same schematic different metrics, such as mean diffusivity and number of fibers derived from DTI data, with regional volume and cortical thickness obtained from T1-w images, and fMRI node degree derived from rs-fMRI data. This approach can lead to better discrimination of groups, and provide an uniform view of different metrics. This connectogram representation is also a comprehensive example of the toolbox capabilities, displaying multimodal imaging and graph-theory metrics, structural connections between brain regions and group statistics.

Limitations and Future Developments

It is very important to understand that several of the methods described throughout this paper still require validation to a certain extent. For the particular case of dMRI, new methods are required to fully reflect the index of anisotropic diffusion in regions of complex white matter configurations like crossing fibers, which cannot be resolved with DTI. Efforts have been made to devise techniques that allow to further explore the structural basis of human neuroanatomy, and these advents might be useful to validate techniques such as tractography (Amunts et al., 2013; Chung et al., 2013). New techniques are also associated with advances in methodological advances which may help us increase the resolution and speed of current techniques, maintaining or even increasing the obtained signal-to-noise ratio (Feinberg & Setsompop, 2013; Setsompop et al., 2012). These techniques are not exclusive for improving structural imaging but also for functional information (Feinberg et al., 2010). Effective connectivity algorithms still lack a gold standard onto which we can compare our results. It is worrying to think that in some cases causality is inferred from data with a small temporal resolution, when the actual neural stimulus is transmitted in the miliseconds time scale (Rodrigues & Andrade, 2014). To overcome this limitation, it is also possible to explore other techniques such as EEG and magnetoencephalography, which provides a much better picture of the dynamic activations of the human brain (Dammers et al., 2007). Finally, in order to efficiently compute large databases of multimodal brain connectivity analysis, the current implementation of the MIBCA toolbox only performs ROI-based analysis and not voxel-by-voxel analysis. Optimization and parallelization of the current algorithms is being pursuit for this purpose and should be available in future releases.

Conclusion

We have introduced a flexible and automatic multimodal approach for the analysis of brain connectivity that can integrate information from different imaging modalities (MRI and PET). While bridging the gap between the high numbers of packages and tools widely available for the neuroimaging community, including pre-processing, connectivity and graph theoretical analyses in one toolbox, MIBCA also offers different possibilities for combining, analysing and visualising data in novel ways enabling a better understanding of the human brain. This is also a request from the neuroimaging community, where the number of multi-modal systems available worldwide (e.g., MR-PET) increased considerably in the last two years.

Supplemental Information

Supplemental Information 1 Multimodal Connectogram (Interactive pdf)

Blue fibers—Direct structural/functional connections; Red fibers—Mediated functional connections.

Click here for additional data file.

Additional Information and Declarations

Competing Interests

Author Contributions

Human Ethics

1 The toolbox is demonstrated with data from the International Consortium for Brain Mapping (ICBM) dataset http://www.loni.usc.edu/ICBM/.

2 The path presented is the one with the fewest number of intermediate regions.

3 The matrices connectivity variance was calculated but not shown in this work.

The authors declare there are no competing interests.

Andre Santos Ribeiro conceived and designed the experiments, performed the experiments, analyzed the data, contributed reagents/materials/analysis tools, wrote the paper, prepared figures and/or tables, reviewed drafts of the paper.

Luis Miguel Lacerda and Hugo Alexandre Ferreira conceived and designed the experiments, performed the experiments, analyzed the data, contributed reagents/materials/analysis tools, wrote the paper, reviewed drafts of the paper.

The following information was supplied relating to ethical approvals (i.e., approving body and any reference numbers):

Data were obtained from the freely available International Consortium for Brain Mapping (ICBM) database: http://www.loni.usc.edu/ICBM/.

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
