# Peer review of "Multimodal Imaging Brain Connectivity Analysis (MIBCA) toolbox"

_PeerJ, doi:10.7717/peerj.1078_

## Round 0.1 · original submission · Major Revisions

As you can read, both reviewers find merit in your work, but much work is needed to upgrade the current paper to a publishable quality. I would suggest you pay attention to how you communicate the content of your toolbox, having in mind a clear readership profile. Please reply to each comment in detail, and highlight the changes so that they are visible to the reviewers.

Reviewer 1 ·

Basic reporting

See the general comments

Experimental design

See the general comments

Validity of the findings

See the general comments

Additional comments

Dear authors,

From the begining of the abstract the software described in, seems very promising. It is very crucial to have all the pre-processing pipelines gathered from various software into one toolbox for a certain purpose. Also it is an advantage that MIBCA is fully automated toolbox for all types of connectivity analysis and for various modalities. It is also nice that MIBCA can visualize all types of graphs, which is curcial if we want to go one step further by investigating directed and weighted graphs. Moreover, what is fascinating is that MIBCA can export statistical results gained from the configuration of the various connectivity matrices and graphs. Summarizing all these I think that this paper should be published after considering the following comments.

Major Comments

Introduction:
It might be worth to introduce connectivity in general, then contrasting briefly micro- and macroscale connectivity regarding pros and cons.
Introducing macroscale more thouroughly would be nice since the toolbox addresses macroscale connectivity specifically
- e.g. what kind of 'resolution' constitutes macroscale?
- What are macroscale measures/techniques are there, e.g. structural MRI, functional MRI, EEG, MEG, PET?
- What do these measures or connectivity derived from these measures reflect / what kind of signals are they, e.g. static or timecourse?
- How do different macroscale measures differ from each other and what aspects of connectivity can they describe?
- Advantages for structure-function relationship, e.g. non-invasive, in-vivo
It might also be nice to highlight the special feature of the toolbox: Lacks arguements for why exactly combining connectivity measures from different imaging modalities is a very important step in advancing the understanding of the connectome.
Also, when differentiating between anatomical, functional, and effective, generally it is referred to as modes or types of connectivity . The term 'level' is more commonly used for making a distinction between micro- and macroscale.

Introduction / Brain Connectivity Analysis Toolboxes:
For a more accessible overview it might be nice to provide the reader with a table comparing the different toolboxes. E.g. columns for different features (functional, antomical, preprocessing, graph analysis etc.) and each line describing a toolbox by putting an 'X' if included

Material and Methods:
Its not clear to me whether this part describes the toolbox in general or the analysis that was used for demonstration purposes.
It would be nice to have some insight into which parameters, features, analysis steps or order are customizable. One example would be: Is it possible to select different atlases for parcellation? Is it also possible to address voxel-by-voxel connectivity instead of ROI-by-ROI. Or regarding graph metircs, are other metrics offered, too?
Visualization: Is it possible to select from different color schemes or to customize it? This would be important considering that the jet color scheme is problemetic due to uncontrolled changes in luminance making certain colors perceptually more prominant than others as well as misleading perceptions of gradients where colors change.

Results:
Most of the results, e.g. the cross-modality comparison of inter- and intrahemispheric connectivity seems to be based on visual inspection of the figures exclusively. This should either be made more explicit or should be backed up by some indices that are statistically tested.

Discussion:
The discussion focuses on the example analysis only. It would benefit from a summarizing paragraph at the beginning (e.g. aim of article, short description of toolbox and most imortant features). It would be nice to have a clear distinction when the discussion refers to the toolbox in general and when it refers to the example analysis.
When describing the example analysis, it would be nice to point out why it would have been harder / not possible / more time consuming / etc. to do this with other toolboxes. What is the special value of using this specific toolbox?


Minor Comments

The paragraph describing the graphs in the 3rd page (lines 57-65) should be shortened.

In Fig.1 the Graph seems to be overlayed to the cortex trying to give a 3D prespective but the result isn’t so good. In order to give a realistic 3D prespective you should somehow give the depth. For example someone could not understand where the nodes are exactly and where the arrows are pointing. According to my opinion you should draw the network using the 3D coordinates of the nodes in the same figure as the cortex and then make the cortex more transparent.

In line 89 you mention that the toolbox for functional connectivity is used in rs-fMRI and in BOLD signals. This toolbox can actually be used in every kind of signals (not only neuroscientific ones) in order to compute their similarity. So it can also be applied in other neuroscientific modalities like M/EEG etc.

Regarding eConnectome I believe that its basic advantage is that it can compute dynamic graphs with extremely high resolution. Their visualization also is gorgeous…

Please rephrase the lines 241-245. I cannot understand what you mean and how this is done…

In the formation of normalized graph indices you should mention how many random graphs you take.


I don’t see any link where the readers can download the MIBCA software. This should be face upon publication.

I propose you to make a website and a mailing list for your software in order to enhance your support in possible future bugs and/or computation problems and questions.
After publication, if it is accepted, I also propose you to advertise it in the various existing maling lists (SPM, EEGLAB, etc.)

Combining data from different modalities effectively would be great, but it's tricky because different imaging modalities produce data that is inherently very different, and comparing/interpreting connectivity measures from the different modalities should be done with caution. This should be acknowledged more clearly.

·

Basic reporting

This paper presents a data analysis toolbox that has a potential to improve the way we analyse multimodal neuroimaging data. However, the manuscript is tragically flawed, by the following reasons:
1. Even though the main topic of the paper is a piece of software the manuscript does not mention how this software can be obtained and used, by the members of the academic community. It is not a simple omission - to access the toolbox I had to ask the PeerJ editors to interfere. In return I was presented with a private repository containing and undocumented code dump.
2. The software lacks any documentation explaining users to to install it or use it.
3. I could not find any tests in the code. This topic was not covered in the manuscript. How can we trust software without unit tests?
4. There is no mention about what license is the code or software are distributed under.
5. Authors do a poor job comparing their toolbox with other connectivity pipelines. They do list many of the existing toolboxes (although they missed: http://journal.frontiersin.org/Journal/10.3389/fninf.2011.00003/full), but they don't clearly show how their new toolkit compares to existing solutions. A table comparing all toolboxes and highlighting the advantages of MIBCA would greatly benefit the manuscript.

It's unclear if the software described in the manuscript is even intended for public consumption - if it's not it should this should be openly stated in the manuscript, but then I see not point in publishing this paper. If it is intended for the general public it is no where ready to be used anyone but the authors themselves. This needs to change for your work to make meaningful impact (which I believe it can).

Experimental design

n/a

Validity of the findings

n/a

Additional comments

I believe your work is valuable and MIBCA has a potential to become a useful tool. However, the way it is presented right now precludes anyone but you to use it. I truly hope this is not your intention and I am looking forward to seeing improvements on the sides of both the software and the manuscripts that would allow the broad academic community to benefit from your work.

---

## Round 0.2 · Minor Revisions

I agree with the positive comments. The revision improved the manuscript considerably. I invite you to meet the last requests for clarification.

Reviewer 1 ·

Basic reporting

see below

Experimental design

see below

Validity of the findings

see below

Additional comments

Dear authors,

Well done! I think that your paper worths to be published in PeerJ. Just a simple notice to take into account prior to publication. You should mention in the Limitation paragraph that the MBICA toolbox cannot perform voxel by voxel analysis.

·

Basic reporting

The following concerns from my initial review were not addressed in the rebuttal or the revised manuscript:

3. I could not find any tests in the code. This topic was not covered in the manuscript. How can we trust software without unit tests?
4. There is no mention about what license the code or software are distributed under.

Experimental design

n/a

Validity of the findings

n/a

Additional comments

n/a

---

## Round 0.3 · accepted · Accept

I congratulate the authors on this submission, which represents a valuable contribution to the field.